# KeyMorph: Robust Multi-modal Affine Registration via Unsupervised Keypoint Detection

**Evan M. Yu**[1]                                                     EMY24@CORNELL.EDU
**Alan Q. Wang**[2]                                                   AW847@CORNELL.EDU
**Adrian V. Dalca**[*3,4]                                             ADALCA@MIT.EDU
**Mert R. Sabuncu**[*2,5]                                           MSABUNCU@CORNELL.EDU

[1] *Nancy E. and Peter C. Meinig School of Biomedical Engineering, Cornell University*

[2] *School of Electrical and Computer Engineering, Cornell University and Cornell Tech*

[3] *Martinos Center for Biomedical Imaging, Massachusetts General Hospital, Harvard Medical School*

[4] *Computer Science and Artificial Intelligence Laboratory (CSAIL), MIT*

[5] *Department of Radiology, Weill Cornell Medical*

**Editors:** Under Review for MIDL 2022

## Abstract

Registration is a fundamental task in medical imaging, and recent machine learning methods have become the state-of-the-art. However, these approaches are often not interpretable, lack robustness to large misalignments, and do not incorporate symmetries of the problem. In this work, we propose KeyMorph, an unsupervised end-to-end learning-based image registration framework that relies on automatically detecting corresponding keypoints. Our core insight is straightforward: matching keypoints between images can be used to obtain the optimal transformation via a differentiable closed-form expression. We use this observation to drive the unsupervised learning of anatomically-consistent keypoints from images. This not only leads to substantially more robust registration but also yields better interpretability, since the keypoints reveal which parts of the image are driving the final alignment. Moreover, KeyMorph can be designed to be equivariant under image translations and/or symmetric with respect to the input image ordering. We demonstrate the proposed framework in solving 3D affine registration of multi-modal brain MRI scans. Remarkably, we show that this strategy leads to consistent keypoints, even across modalities. We demonstrate registration accuracy that surpasses current state-of-the-art methods, especially in the context of large displacements. Our code is available at https://github.com/evanmy/keymorph

**Keywords:** Image registration, Multi-modal, Keypoint detection, Unsupervised Learning

## 1. Introduction

Registration is a core problem in biomedical imaging applications. Multiple images, often encompassing a variety of contrasts, are commonly acquired (Uludağ and Roebroeck, 2014). Classical (i.e. non-learning-based) registration methods involve an iterative optimization of a similarity metric over a space of transformations (Oliveira and Tavares, 2014; Sotiras et al., 2013). Recent deep learning-based strategies leverage large datasets of images to solve registration. Given a pair of images $(\boldsymbol{x}_f, \boldsymbol{x}_m)$, these strategies use neural network architectures that either output transformation parameters (e.g. affine or spline) (Lee et al.,

---

[*] Joint senior author

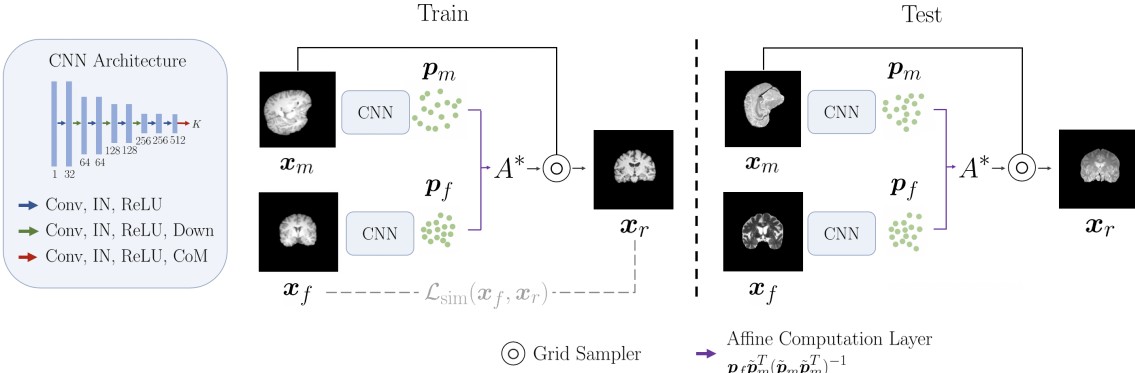

Figure 1: Proposed framework. Fixed and moving 3D images are passed through the same keypoint detection network composed of convolutional layers and a final center-of-mass (CoM) layer that predicts keypoints useful for registration. The affine matrix is then computed using Eq. 2 and is used to transform the moving image.

2019a; de Vos et al., 2019) or a dense deformation field (Balakrishnan et al., 2019) which aligns them. Since the registration step is achieved via a single feed-forward pass, it is substantially faster than iterative methods.

However, prior learning-based methods often fail when the given image pair has a large misalignment. Existing systems typically require that image pairs are roughly aligned (Balakrishnan et al., 2019; Dalca et al., 2019; de Vos et al., 2019; Qin et al., 2019), or at least in the same orientation (Mocanu et al., 2021). In addition, today's learning-based registration methods lack interpretability, as they are essentially "black-box models" that output either transformation parameters or a deformation field and provide little insight into what drives the alignment. Finally, the machine learning models used in registration tasks generally do not exploit the symmetries and equivariances present in the problem. For example, if one of the images is translated by a fixed amount, this has a pre-determined effect on the optimal registration solution, and this property is not built into any of the architectures used today for image registration.

**Contribution.** We propose KeyMorph, an unsupervised end-to-end deep-learning-based framework that aims to address all aforementioned issues. Our main insight is that *matched keypoints* can be used to derive the optimal transformation in closed-form. This keypoint-based formulation is robust against misalignments as the closed-form solution does not depend on the initial position of the keypoints. Additionally, the model is interpretable since the keypoints that drive the alignment can be visualized. Finally, we also show how to incorporate symmetries into the model design. For example, the architecture we describe in this paper is translation equivariant and leverages a recently-proposed center-of-mass layer (Ma et al., 2020; Sofka et al., 2017). Rather than treating matched keypoint detection as a supervised learning problem requiring human-annotated keypoints, we propose to use an end-to-end unsupervised strategy tailored toward registration. By unifying image registration and keypoint detection, we can train a model that finds matching keypoints useful

for aligning images. We demonstrate this framework in the context of affine registration of 3D multi-modal brain MR scans.

## 2. Related Works

**Classical Image Registration Methods.** Pairwise iterative, optimization-based approaches have been extensively studied in medical image registration (Hill et al., 2001; Oliveira and Tavares, 2014). These methods employ a variety of similarity functions, types of deformation, transformation constraints or regularization strategies, and optimization techniques. Intensity-based similarity criteria are most often used, such as mean-squared error (MSE) or normalized cross correlation for registering images of the same modality (Avants et al., 2009, 2008; Hermosillo et al., 2002). For registering image pairs from different modalities, statistical measures like mutual information or contrast-invariant features like MIND are popular (Heinrich et al., 2012; Hermosillo et al., 2002; Hoffmann et al., 2020; Mattes et al., 2003; Viola and Wells III, 1997).

Another registration paradigm first detects features or keypoints in the images, and then establishes their correspondence (Myronenko and Song, 2010). This approach often involves handcrafted features (Tuytelaars and Mikolajczyk, 2008), features extracted from curvature of contours (Rosenfeld and Thurston, 1971), image intensity (Förstner and Gülch, 1987; Harris et al., 1988), color information (Montesinos et al., 1998; Van de Weijer et al., 2005), or segmented regions (Matas et al., 2004; Wachinger et al., 2018). Features can be also obtained so that they are invariant to viewpoints (Bay et al., 2006; Brown et al., 2005; Lowe, 2004; Toews et al., 2013). These algorithms then optimize similarity functions based on these features over the space of transformations (Chui and Rangarajan, 2003; Hill et al., 2001). This strategy is sensitive to the quality of the keypoints and often suffer in the presence of substantial contrast and/or color variation (Verdie et al., 2015).

**Learning-based Methods.** In learning-based image registration, supervision can be provided through ground-truth transformations, either synthesized or computed by classical methods (Cao et al., 2018; Dosovitskiy et al., 2015; Eppenhof and Pluim, 2018; Lee et al., 2019b; Uzunova et al., 2017; Yang et al., 2017). Unsupervised strategies use loss functions similar to those employed in classical methods (Balakrishnan et al., 2019; Dalca et al., 2019; de Vos et al., 2019; Fan et al., 2018; Krebs et al., 2019; Qin et al., 2019; Wu et al., 2015). Weakly supervised models employ (additional) landmarks or labels to guide training (Balakrishnan et al., 2019; Fan et al., 2019; Hu et al., 2018a,b).

Recent learning-based methods compute image features or keypoints (Ma et al., 2021) that can be used for image recognition, retrieval, or registration. Learning the keypoints can be done with supervision (Verdie et al., 2015; Yi et al., 2016), self-supervision (DeTone et al., 2018) or without supervision (Barroso-Laguna et al., 2019; Lenc and Vedaldi, 2016; Ono et al., 2018). The goals and problems that the cited computer vision papers tackle are different from ours. In contrast, our focus is on robust image registration via corresponding keypoints; these prior works aim explicitly to obtain keypoints that are repeatable under different viewpoints and/or image acquisition conditions. Nevertheless, we build on previous ideas and introduce a framework that output matched keypoints in 3D space regardless of the initial position of the image.

## 3. Proposed Method

Let $\boldsymbol{x}_m$ and $\boldsymbol{x}_f$ be moving (source) and fixed (target) volumes, which may vary in modality and orientation.[1] In this paper, we focus on 3D affine transformations, where the goal is to find the optimal affine transformation matrix $A^* \in \mathbb{R}^{3 \times 4}$ such that the moved (registered) image $\boldsymbol{x}_r = \boldsymbol{x}_m \circ A^*$ matches the fixed image $\boldsymbol{x}_f$, where $\circ$ denotes the spatial transformation of an image. To efficiently compute $A^*$, we employ *corresponding* keypoints from $\boldsymbol{x}_m$ and $\boldsymbol{x}_f$. The matching keypoints are computed using a convolutional neural network, and $A^*$ is estimated using a non-learnable computation layer.

### 3.1. Keypoint Detector Network

To compute $K$ matching keypoints, we use a single neural network $g_\phi$ with parameters $\phi$ to produce $\boldsymbol{p}_m = g_\phi(\boldsymbol{x}_m)$ and $\boldsymbol{p}_f = g_\phi(\boldsymbol{x}_f)$, where $\boldsymbol{p}_m$ and $\boldsymbol{p}_f$ are matrices of shape $d \times K$ (i.e. each column is a corresponding keypoint pair of $d$ dimensions).

In our implementation, the backbone architecture of the keypoint detector consists of convolutional layers, followed by instance normalization (Ulyanov et al., 2016), ReLU activation, and 2x downsampling via strided convolution, as shown in Fig. 1. The output from the backbone is followed by a center-of-mass (CoM) layer (Ma et al., 2020; Sofka et al., 2017), which computes the center-of-mass for each of the $K$ activation maps. This specialized layer is (approximately) translation equivariant and enables precise localization. We provide more details and compare CoM to fully connected layers in Appendix B.

### 3.2. Affine Computation Layer

Given $K$ corresponding keypoint pairs, we can derive a differentiable closed-form expression for an affine transformation that aligns the keypoints. Let $\tilde{\boldsymbol{p}}_m = [\boldsymbol{p}_m\ \mathbb{1}]^T \in \mathbb{R}^{(d+1) \times K}$, where $\mathbb{1} \in \mathbb{R}^{1 \times K}$ is a vector of ones and $K > d$. We aim to find the optimal affine transformation

$$A^* = \arg\min_A \|A\tilde{\boldsymbol{p}}_m - \boldsymbol{p}_f\|_F, \tag{1}$$

where $\|\cdot\|_F$ denotes the Frobenius norm. This leads to the closed-form solution

$$A^* = \mathcal{A}(\boldsymbol{p}_f, \boldsymbol{p}_m) = \boldsymbol{p}_f \tilde{\boldsymbol{p}}_m^T (\tilde{\boldsymbol{p}}_m \tilde{\boldsymbol{p}}_m^T)^{-1}. \tag{2}$$

We provide the derivation in the Appendix A.

### 3.3. Training

We found the following self-supervised pre-training strategy to be effective for initializing the keypoint detector backbone. We first pick a set of initial *keypoints* $\boldsymbol{p}$ chosen uniformly at random over the image coordinate grid. For a single subject, we assume that we have access to aligned multi-modal (e.g. T1, T2, PD-weighted MRI) volumes $\{\boldsymbol{x}_i\}$. During

---

1. Although we consider 3D volumes in this work, we stress that our method is agnostic to the number of dimensions. The terms "image" and "volume" are used interchangeably.

pre-training, in each mini-batch, we apply random affine transformations $\mathcal{T}$ (drawn from a uniform distribution over the parameter space) to $\boldsymbol{x}_i$ and $\boldsymbol{p}$, and minimize:

$$\arg\min_{\phi} \sum_i \mathbb{E}_{\mathcal{T}} \left\| \mathcal{T}\boldsymbol{p} - g_\phi(\boldsymbol{x}_i \circ \mathcal{T}) \right\|_2^2, \tag{3}$$

where $\mathbb{E}$ denotes expectation and $\mathcal{T}\boldsymbol{p}$ transforms the list of coordinates in $\boldsymbol{p}$ by the affine transformation $\mathcal{T}$.

Following the pre-training strategy, we train KeyMorph on random image pairs using the entire training dataset. We consider the typical real-world scenario of multi-parametric MRI, where scans of potentially different contrasts (e.g. T1, T2, PD-weighted) need to be registered. We present only within-modality image pairs to the network during training, where a simple loss like MSE may be used. Thus, given a dataset $\mathcal{D}$ composed of same-modality image pairs $(\boldsymbol{x}_f, \boldsymbol{x}_m)$, the overall objective is:

$$\arg\min_{\phi} \mathbb{E}_{(\boldsymbol{x}_f, \boldsymbol{x}_m) \sim \mathcal{D}} \left\| \boldsymbol{x}_m \circ \mathcal{A}\left(g_\phi(\boldsymbol{x}_f), g_\phi(\boldsymbol{x}_m)\right) - \boldsymbol{x}_f \right\|_2^2. \tag{4}$$

During training, we apply random affine transformations to the images as an augmentation strategy. Remarkably, we found that KeyMorph learns to detect anatomically consistent keypoints *across modalities*, even though it is trained on same-modality pairs. In our experiments, we demonstrate how this can be used to perform multi-modal registration at test-time.

### 3.4. KeyMorph Variants

We use `KeyMorph_mse` to refer to the main unsupervised variant of our model, trained with Equation 4. In addition, we employ a *supervised* variant, `KeyMorph_dice`, that exploits segmentations during training. During pre-training of this variant, we use the center-of-mass of segmentation labels as ground truth keypoints and do not restrict to a single subject. During subsequent end-to-end training, we use soft-Dice, a loss function that measures volume overlap of the moving and registered label maps (Balakrishnan et al., 2019; Hoffmann et al., 2020; Hu et al., 2018a).

## 4. Experiments

### 4.1. Dataset

We used the IXI brain MRI dataset[2] for evaluation. Each subject has T1, T2, and PD-weighted 3D MRI scans in spatial alignment. We partitioned the 577 total subjects into sets of 427, 50, and 100 for training, validation, and testing, respectively. We performed standard skull stripping (Kleesiek et al., 2016) on all images.

We used a pre-trained and validated SynthSeg model (Billot et al., 2020) to automatically delineate 23 regions of interest (ROIs)[3]. We used these segmentations for training a subset of models, as described below. Furthermore, all performance evaluations were based on examining the overlap of ROIs in the test images.

---

2. https://brain-development.org/ixi-dataset/

3. ROIs were pallidum, amygdala, caudate, cerebral cortex, hippocampus, thalamus, putamen, white matter, cerebellar cortex, ventricle, cerebral white matter, and brainstem.

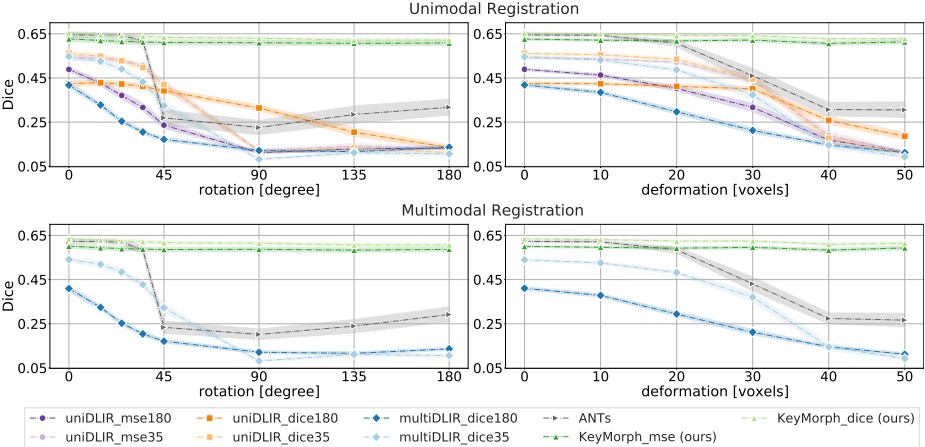

Figure 2: **Registration results.** The x-axis of the second column shows the average absolute displacement in the moving volume after rotation, scaling, or translation. The Dice score is averaged for all test subjects and brain anatomical regions. See Section 4.3 for details on the naming scheme.

## 4.2. Test-time Performance Evaluation

We used each testing subject as a moving volume $\boldsymbol{x}_m$, paired with a different test volume treated as fixed image $\boldsymbol{x}_f$. For all test volumes, we use the 23-label segmentation maps to quantify alignment. We simulated different degrees of misalignment by transforming $\boldsymbol{x}_m$ using rotation, scaling, or translation. For a given transformation type, we choose 1-3 axes randomly and apply a uniform random transformation (e.g. rotation) up to a given amount (e.g. degree). We use the predicted transformation to resample the moved segmentation labels on the fixed image grid, and compute the Dice score to quantify alignment quality.

We performed registration across all combinations of available modalities (registering T1 to T1, T1 to T2, etc). All pairings and amount of transformations was kept the same across the registration experiments.

## 4.3. Baselines

**Advanced Normalizing Tools (ANTs)** is a widely used software package for medical image registration (Avants et al., 2009). We use the "TRSAA" affine implementation, which consist of translation, rigid, similarity and two affine transformation steps. The volumes are registered successively at three different resolutions: 0.25x, 0.5x and finally at full resolution. At 0.25x and 0.5x resolution, Gaussian smoothing with $\sigma$ of two and one voxels is applied, respectively. In addition, we use the ANTs affine initializer, which conducts a grid search over a range of rotations and translations to find a good initialization. We used mutual information as the similarity metric, which is suitable for registering images with different contrasts.

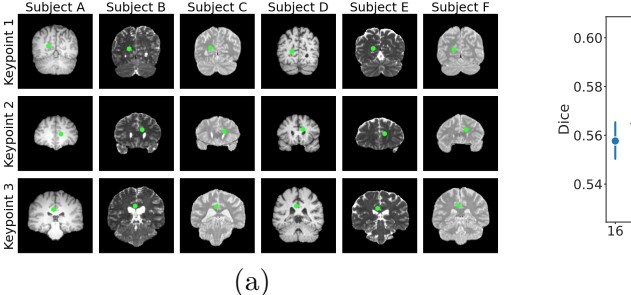

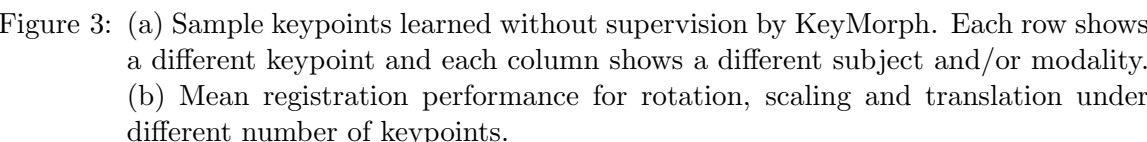

(a)           (b)

Figure 3: (a) Sample keypoints learned without supervision by KeyMorph. Each row shows a different keypoint and each column shows a different subject and/or modality. (b) Mean registration performance for rotation, scaling and translation under different number of keypoints.

**Deep Learning for Image Registration (DLIR)** is a recent learning-based method that includes affine image registration (de Vos et al., 2019). DLIR implements a Spatial Transformer Network (Jaderberg et al., 2015), which was originally used to improve class prediction accuracy and has since become the backbone of many subsequent works in image registration (de Vos et al., 2017; Lee et al., 2019a; Balakrishnan et al., 2019). For a direct comparison, we used the same backbone architecture as KeyMorph replacing the center-of-mass layer with a fully-connected (FC) layer which outputs 12 parameters for the 3D affine transformation.[4] In Appendix B, we also investigated a KeyMorph implementation with the same FC layer as DLIR.

We find that DLIR often cannot register image pairs with large misalignments. We alleviate this by using more aggressive augmentation during training, where the images are randomly transformed. We consider two different amounts of rotation for the training of DLIR: maximum $\pm 35°$ or $\pm 180°$. We use the same loss function and training scheme as we used for KeyMorph. We trained separate DLIR models for each modality as it produces better results than training DLIR across modalities with mutual information. We also trained *supervised* modality-specific and multi-modal DLIR models using a soft-Dice loss computed on the aligned segmentation maps (Lee et al., 2019b).

Altogether, we implement six different DLIR variants. The naming scheme follows the convention `<mod>DLIR_<loss><degree>`, where `<mod>` denotes whether the model was trained on a single (`uni`) or multiple (`multi`) modalities, `<loss>` denotes the training loss function, and `<degree>` denotes the maximum angle of rotation for augmentation.

## 5. Results

Fig. 2, Table A.2 and Appendix F summarize the results. We find that all DLIR models suffer substantially as the rotation angle increases. Training with aggressive augmentation increases performance for test pairs with large misalignments, but reduces the accuracy for those with smaller misalignments. Using supervision (`uniDLIR_dice`) leads to improved

---

4. We used instance Norms for `multiDLIR` and Batch Norms for `uniDLIR`, which we found to work well in practice.

accuracy. For unimodal registration, the DLIR model that was trained with all modalities (`multiDLIR_dice`) did not produce better accuracy than a model that was trained with each modality separately. ANTs yields excellent results when the initial misalignment is small (e.g., less than 45 degrees rotation). However, the accuracy drops substantially when the misalignment exceeds this range.

In contrast to these models, KeyMorph variants performed well across all types of transformations and ranges, with only marginal drops in accuracy in large misalignments. In the case where we have access to ROIs during training, we find that KeyMorph trained with Dice outperforms all models in nearly all the tasks. However, the unsupervised variant trained with MSE still yields excellent accuracy across all settings and is only minimally suboptimal compared to its supervised counterpart. We provide qualitative results in Appendix D, and compare the computational time across different models in Appendix C. Overall, the KeyMorph variants substantially outperform other learning-based baselines, and KeyMorph performs comparably or often substantially better (at large misalignments) than state-of-the-art ANTs registration, while requiring substantially less runtime.

### 5.1. Keypoint Analysis

**Number of Keypoints.** We trained four unsupervised KeyMorph model variants with 16, 32, 64, and 128 keypoints, respectively. Fig. 3b illustrates that increasing the number of keypoints leads to improved Dice scores (and lower variability), up to 64 keypoints. However, we find that further increasing the keypoints can lead to a drop in performance (at 128 keypoints). We hypothesize that it is harder to find a relatively large number of anatomical landmarks that are consistent across individuals.

**Keypoint Visualization.** In contrast to existing models that compute the transformation parameters using a "black-box" neural network, we can investigate the keypoints that KeyMorph learns to drive the alignment. Fig. 3a illustrates a set of representative keypoint examples for a learned unsupervised KeyMorph model. We find that the final learned keypoints correspond to the same anatomical region in different subjects and modalities. In appendix G, we provide a quantitative study on keypoint consistency across the scans of each subject, and in Appendix H we show an expanded version of Fig. 3a.

## 6. Conclusion

We introduce KeyMorph, a robust deep learning-based affine registration framework that employs unsupervised keypoint extraction. Our key insight is that matched keypoints yield a closed-form solution for affine registration, even in the case of large misalignments, and this in turn can be used to drive unsupervised keypoint detection. We showed that state-of-the-art optimization and learning-based methods for image registration struggle to register image pairs that have large misalignment. In contrast to many "black-box" machine learning-based registration methods, KeyMorph also offers the ability to investigate what drives the registration by visualizing the keypoints. We envision that KeyMorph can be used in a variety of applications that exhibit large misalignments, and can be extended to compute non-linear deformations.

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

## Appendix A. Derivation of Closed-form Expression

Let $\boldsymbol{p}_f \in \mathbb{R}^{d \times K}$ and $\tilde{\boldsymbol{p}}_m = [\boldsymbol{p}_m\ \mathbb{1}]^T \in \mathbb{R}^{(d+1) \times K}$, where $\mathbb{1} \in \mathbb{R}^{1 \times K}$ is a vector of ones and $K > d$. We wish to find the optimal affine transformation $A \in \mathbb{R}^{d \times (d+1)}$ which minimizes:

$$\mathcal{L} = \|A\tilde{\boldsymbol{p}}_m - \boldsymbol{p}_f\|_F \,,$$

where $\|\cdot\|_F$ denotes the Frobenius norm. Taking the derivative with respect to $A$ and setting the result to zero, we obtain:

$$\begin{aligned}
\frac{\partial \mathcal{L}}{\partial A} &= (A\tilde{\boldsymbol{p}}_m - \boldsymbol{p}_f)\tilde{\boldsymbol{p}}_m^T = \boldsymbol{0} \\
&\implies A\tilde{\boldsymbol{p}}_m\tilde{\boldsymbol{p}}_m^T = \boldsymbol{p}_f\tilde{\boldsymbol{p}}_m^T \\
&\implies A = \boldsymbol{p}_f\tilde{\boldsymbol{p}}_m^T(\tilde{\boldsymbol{p}}_m\tilde{\boldsymbol{p}}_m^T)^{-1}.
\end{aligned}$$

## Appendix B. Center-of-Mass Layer vs Fully Connected Layer

The backbone of our architecture is composed of CNN layers and a center-of-mass (CoM) layer at the end of the network (Ma et al., 2020; Sofka et al., 2017). The CoM layer computes the center-of-mass of the activation map for each channel of the CNN output (i.e., the weighted average between the voxel values and the grid coordinates). These center-of-masses are then used as keypoints in KeyMorph. CoM layer is shift-equivariant as compared to the fully connected (FC) layer. As a comparison, we compared the performance of using a CoM layer and FC layer, which is commonly used in registration (de Vos et al., 2019; Jaderberg et al., 2015; de Vos et al., 2017; Lee et al., 2019a). In Fig. A.1, the model using FC layers follows the same architecture as DLIR model that was used in the baseline. However, instead of outputting 12 affine parameters, it outputs 64 keypoints. We repeated the experiments found in Section 4. We can see that regardless of the type of layer used

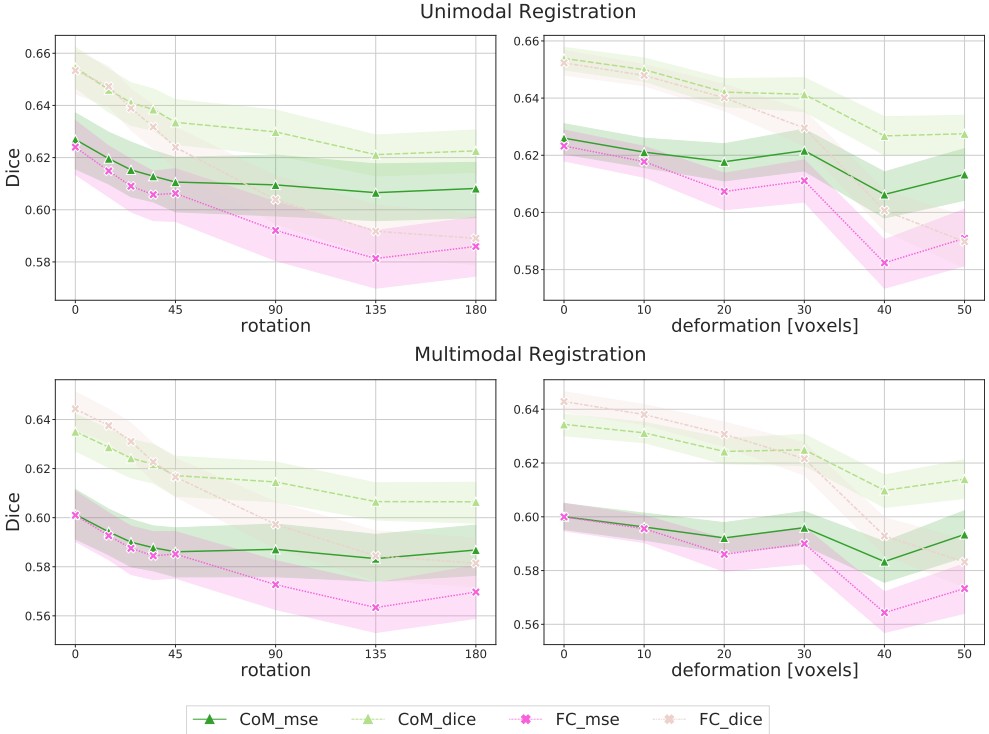

Figure A.1: Performance comparison between KeyMorph models that uses center-of-mass (CoM) and fully connected (FC) layers to predict keypoints. The suffix `mse` and `dice` represent the unsupervised and supervised version of KeyMorph, respectively.

to compute the keypoints, KeyMorph provide robustness to large deformation. Models that uses CoM have comparable performance at low deformation compared to models that uses FC layers. However, CoM models provide the best performance at higher degrees of misalignment.

## Appendix C. Computation Time

DLIR models and KeyMorph took about 3.5 days to train. Table A.1 summarizes runtime at test time of all methods. On a modern CPU, the ANTs baseline, which does not have GPU support, requires more than 60 seconds, whereas KeyMorph requires roughly 2.7 seconds per subject at test time (about 0.2 sec on GPU). Comparing ANTs and KeyMorph (CPU) and KeyMorph (GPU), this represent more than 20x and 300x speed-up, respectively.

## Appendix D. Qualitative Results

We present some qualitative results of KeyMorph trained without supervision. Moving and fixed subjects were picked randomly from the test set. We introduced random affine transformation with $\pm 180$ degrees of rotation, $\pm 20\%$ of scaling, and $\pm 20$ voxels of translation

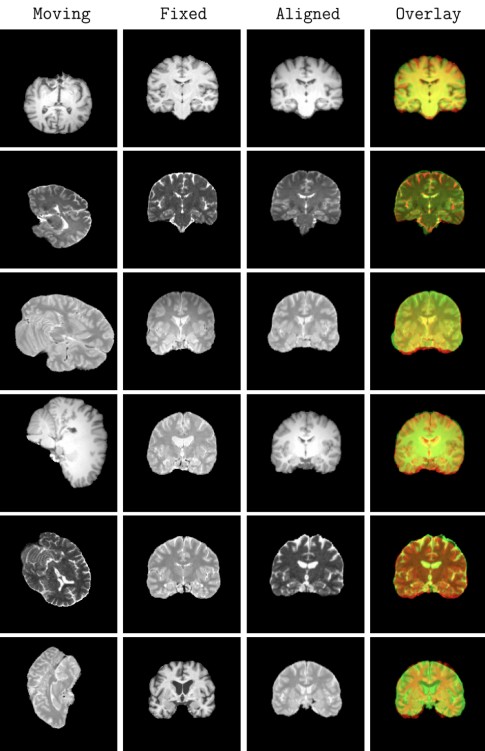

Figure A.2: Sample registration results obtained with KeyMorph trained with no supervision (`KeyMorph_mse`). Each row shows a different moving and fixed image pair. Red is the fixed image and green is resampled (moved) image.

to the moving image. Fig. A.2 presents the registration result of the moved brains in the third column. We overlayed the aligned image (green) to the fixed/target brain (red).

## Appendix E. Quantitative of Results

This section provides more details on the quantitative results for Fig. 2 from the main paper. We randomly picked a different fixed volume $x_f$ on the test set for each moving subject. We introduce a random amount of misalignment to $x_m$. The misalignment include random affine transformation with $\pm 180$ degrees of rotation, $\pm 20\%$ of scaling, and $\pm 20$ voxels of translation. Each of the 3 axes of $x_m$ had 50% chance of being deformed with a

| Model | CPU Time (s) | GPU Time (s) |
|---|---|---|
| ANTs | 66.95±1.56 | - |
| DLIR | 1.49±0.09 | 0.02±0.001 |
| KeyMorph | 2.68±0.44 | 0.21±0.012 |

Table A.1: Average computation time across different models during testing

| T | Model | Dice Score | | | | | | | | |
|---|---|---|---|---|---|---|---|---|---|---|
| | | T1→T1 | T2→T1 | PD→T1 | T1→T2 | T2→T2 | PD→T2 | T1→PD | T2→PD | PD→PD |
| rotation | uniDLIR_mse180 | 26.8±14.84 | - | - | - | 26.86±15.01 | - | - | - | 29.36±17.01 |
| | uniDLIR_mse35 | 51.33±7.95 | - | - | - | 53.88±7.2 | - | - | - | 53.57±6.35 |
| | uniDLIR_dice180 | 38.64±14.65 | - | - | - | 28.59±12.62 | - | - | - | 35.24±13.98 |
| | uniDLIR_dice35 | 52.57±7.81 | - | - | - | 53.41±9.33 | - | - | - | 54.14±8.03 |
| | multiDLIR_dice180 | 21.68±11.06 | 21.78±11.42 | 22.0±11.46 | 21.32±10.6 | 21.88±11.48 | 21.99±11.37 | 21.22±10.67 | 21.78±11.5 | 22.09±11.62 |
| | multiDLIR_dice35 | 49.84±8.48 | 48.98±8.16 | 49.37±8.41 | 49.27±8.33 | 49.42±8.21 | 49.41±8.36 | 49.5±8.3 | 49.15±8.11 | 50.28±8.6 |
| | ANTs | 48.07±26.14 | 44.32±25.34 | 42.27±25.0 | 43.48±25.23 | 45.06±25.59 | 42.69±25.04 | 40.56±25.01 | 43.51±24.89 | 44.66±25.84 |
| | KeyMorph_mse | 62.24±6.56 | 58.75±5.98 | 58.8±6.59 | 58.62±6.07 | 60.79±6.28 | 59.38±5.88 | 58.83±6.01 | 59.37±5.68 | 61.79±6.21 |
| | KeyMorph_dice | **65.17 ± 5.11** | **61.94 ± 5.04** | **62.1 ± 4.82** | **62.54 ± 4.96** | **62.96 ± 5.3** | **61.61 ± 4.83** | **62.15 ± 4.87** | **61.35 ± 5.11** | **63.24 ± 5.26** |
| scaling | uniDLIR_mse180 | 45.97±8.07 | - | - | - | 45.96±8.46 | - | - | - | 53.47±7.32 |
| | uniDLIR_mse35 | 53.32±7.26 | - | - | - | 54.46±7.37 | - | - | - | 53.37±5.98 |
| | uniDLIR_dice180 | 44.23±9.13 | - | - | - | 39.09±7.48 | - | - | - | 42.91±7.74 |
| | uniDLIR_dice35 | 55.19±6.95 | - | - | - | 54.92±7.96 | - | - | - | 57.57±6.8 |
| | multiDLIR_dice180 | 39.12±9.49 | 41.08±7.93 | 41.32±7.81 | 38.12±9.45 | 41.74±8.43 | 41.7±8.1 | 37.96±9.93 | 41.48±8.85 | 42.02±8.81 |
| | multiDLIR_dice35 | 53.43±6.35 | 52.68±6.33 | 52.86±6.54 | 52.57±6.56 | 52.84±6.98 | 52.49±6.71 | 52.69±6.37 | 52.38±6.76 | 53.32±6.74 |
| | ANTs | 66.29±6.19 | **63.4 ± 5.7** | 62.51±5.93 | 62.7±5.81 | 63.84±6.33 | 62.42±5.74 | 61.49±5.94 | 62.28±5.7 | 64.29±6.48 |
| | KeyMorph_mse | 63.0±6.79 | 59.23±6.02 | 59.7±6.65 | 59.35±6.13 | 61.73±6.49 | 60.27±6.01 | 59.66±6.28 | 60.16±5.62 | 62.8±6.64 |
| | KeyMorph_dice | **66.51 ± 5.24** | 63.25±4.86 | **63.35 ± 4.6** | **63.33 ± 4.98** | **64.16 ± 5.35** | **62.7 ± 4.55** | **63.37 ± 5.0** | **62.77 ± 4.81** | **64.85 ± 5.42** |
| translation | uniDLIR_mse180 | 45.88±8.12 | - | - | - | 45.57±8.25 | - | - | - | 53.14±7.73 |
| | uniDLIR_mse35 | 52.8±7.84 | - | - | - | 54.43±7.56 | - | - | - | 53.32±6.33 |
| | uniDLIR_dice180 | 45.05±8.84 | - | - | - | 38.59±7.28 | - | - | - | 43.09±8.07 |
| | uniDLIR_dice35 | 55.26±6.97 | - | - | - | 54.77±7.97 | - | - | - | 57.65±6.69 |
| | multiDLIR_dice180 | 37.52±9.69 | 38.78±8.52 | 39.25±8.35 | 36.32±9.57 | 39.1±9.18 | 39.3±8.79 | 36.18±10.03 | 38.92±9.5 | 39.7±9.38 |
| | multiDLIR_dice35 | 53.41±6.17 | 52.38±6.07 | 52.67±6.5 | 52.84±6.27 | 52.78±6.58 | 52.6±6.68 | 52.88±6.49 | 52.33±6.62 | 53.33±6.93 |
| | ANTs | 66.34±6.37 | 63.45±5.7 | 62.49±5.9 | 62.79±5.81 | 63.94±6.46 | 62.44±5.73 | 61.5±5.97 | 62.34±5.72 | 64.3±6.62 |
| | KeyMorph_mse | 63.5±7.09 | 59.5±5.92 | 59.84±6.64 | 59.86±6.02 | 62.33±6.63 | 60.58±5.9 | 60.19±6.19 | 60.64±5.5 | 63.28±6.78 |
| | KeyMorph_dice | **66.99 ± 5.41** | **63.55 ± 4.82** | **63.75 ± 4.64** | **63.94 ± 4.71** | **64.79 ± 5.62** | **63.34 ± 4.55** | **63.83 ± 4.74** | **63.16 ± 4.86** | **65.42 ± 5.66** |

Table A.2: Mean performance of all method with their standard deviation. The average Dice score is computed across test subject pairs, brain regions, and modalities. The notation $A \to B$ refers to registering moving volumes of modality $A$ to fixed volumes of modality $B$. Bold numbers highlight the highest Dice score of a task given a transformation shown in the first column T.

given transformation with at least one axis being perturbed. Since each subject in the IXI dataset has a corresponding T1, T2 and PD-weighted MRI scan, we performed registration across all combination of modalities (e.g. registering T1 to T1, T1 to T2, etc..). All pairings and random transformations were kept the same across the registration experiments.

## Appendix F. Landmarks Alignment

We repeated the experiment in Section 5. Instead of computing the dice score between different regions of interest, we calculated the root mean square error (rmse) between the center-of-masses of the delineated regions.

## Appendix G. Keypoint Consistency

Keypoint consistency plays an important role for multimodal registration. In this section, we show that KeyMorph learns consistent keypoints across different modalities in an unsupervised manner. In Fig. A.4, we plotted the mean absolute error between the keypoints of each test subject across different modalities as a function of training iteration. We observe that as the model trains, the keypoints become more consistent across modalities. In other words, the keypoints lie in almost the same location across modalities for a given subject, allowing KeyMorph to perform accurate multimodal registration.

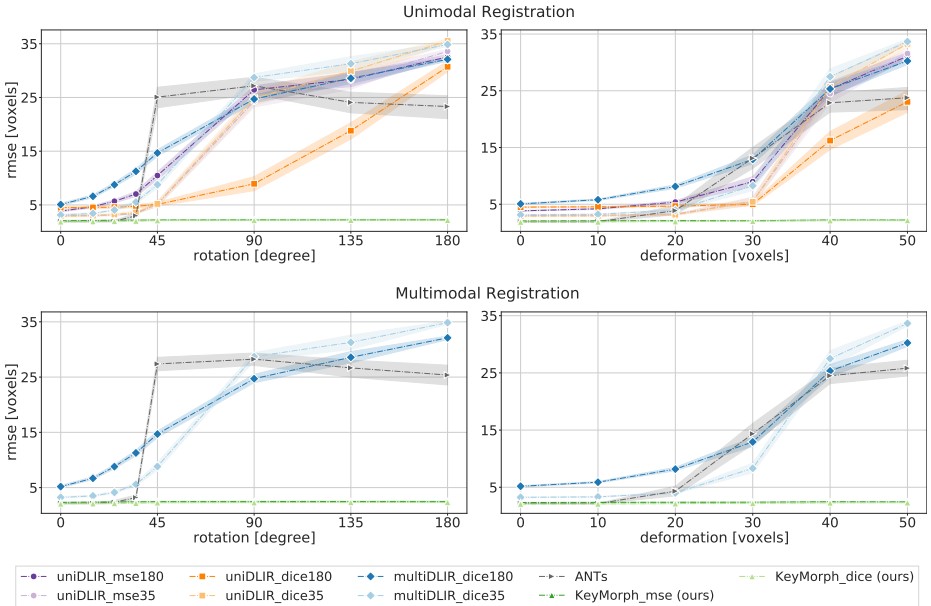

Figure A.3: **Registration results.** The x-axis of the second column shows the average absolute displacement in the moving volume after rotation, scaling, or translation. The rmse is averaged for all test subjects and center-of-masses for each region of interest. See Section 4.3 for details on the naming scheme.

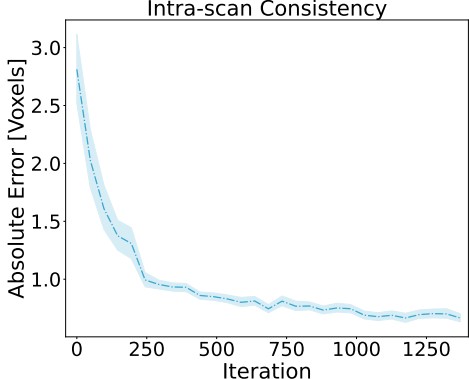

Figure A.4: Mean absolute error between keypoint locations of test subject across different modalities. Each training iteration represent a model update after 32 training subjects.

## Appendix H. Keypoint Visualization

In this section, we provide an extended visualization of keypoints. In the figures below, we investigated which regions KeyMorph uses to register the volume pairs. We picked 12

random subjects and showed 12 of the 64 learned keypoints in each row. These keypoints were learned without supervision of any labeled regions. We can observe that the final keypoints lie within similar region of the brain across different subjects and scans.

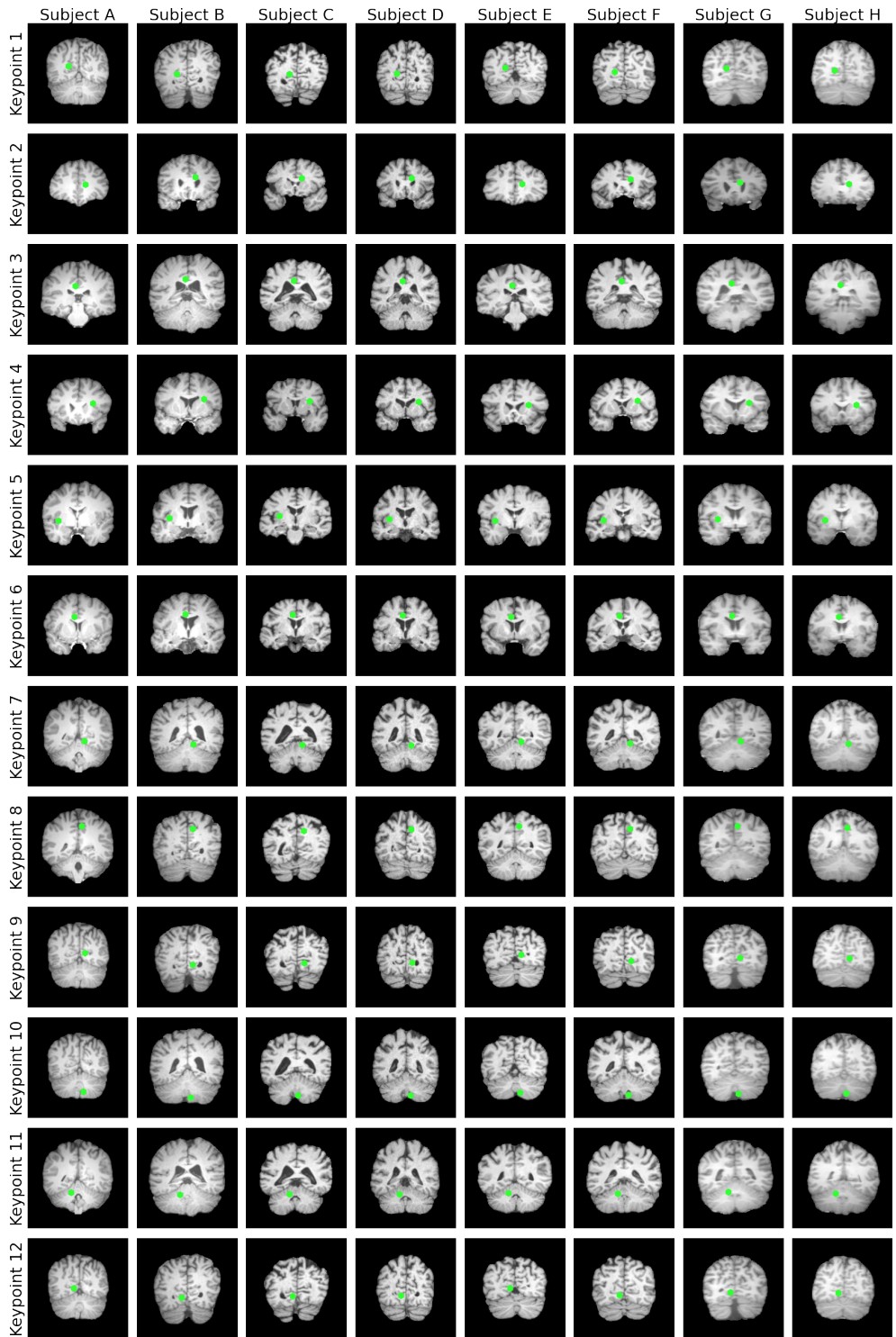

Figure A.5: Keypoints for different T1 scans

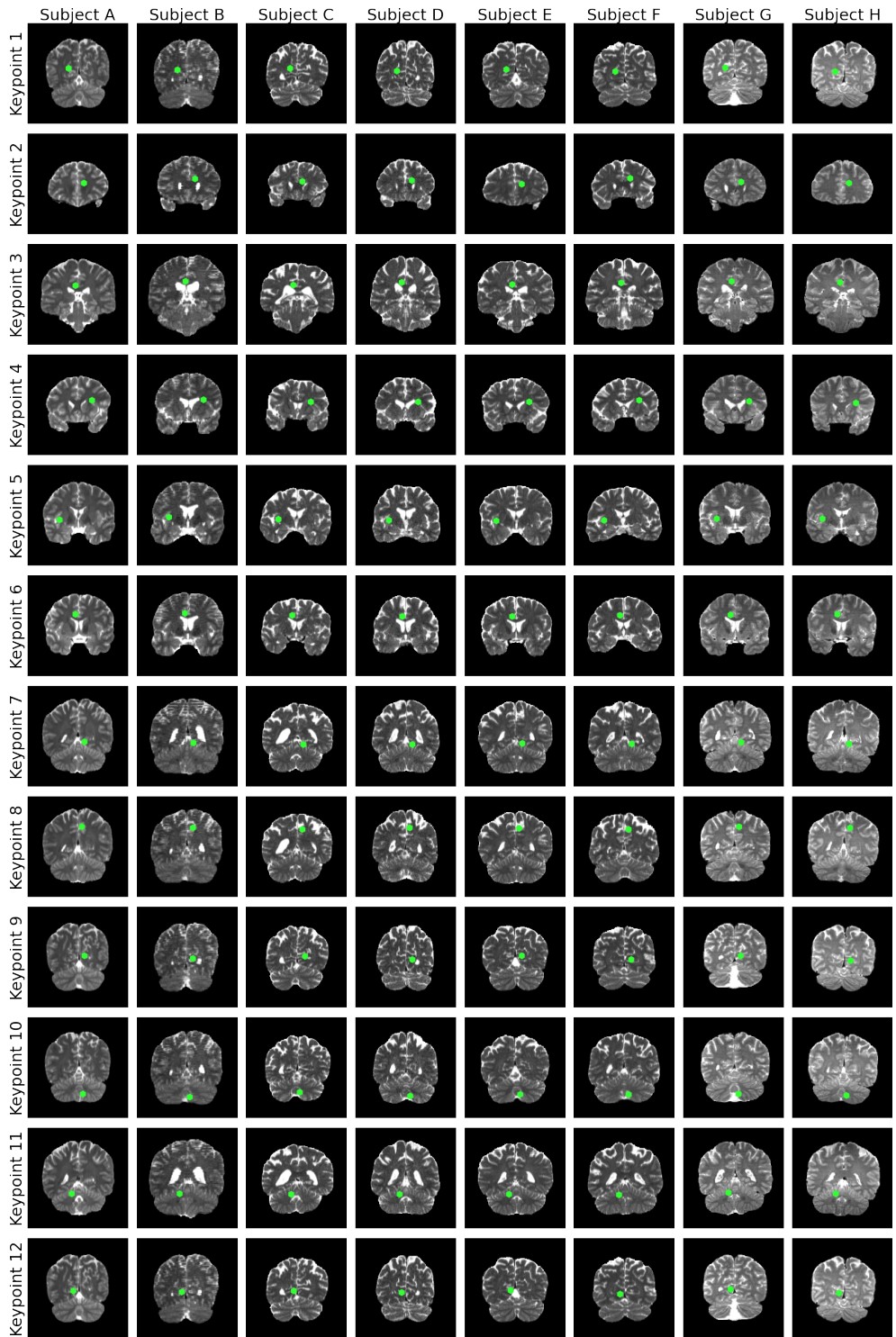

Figure A.6: Keypoints for different T2 scans

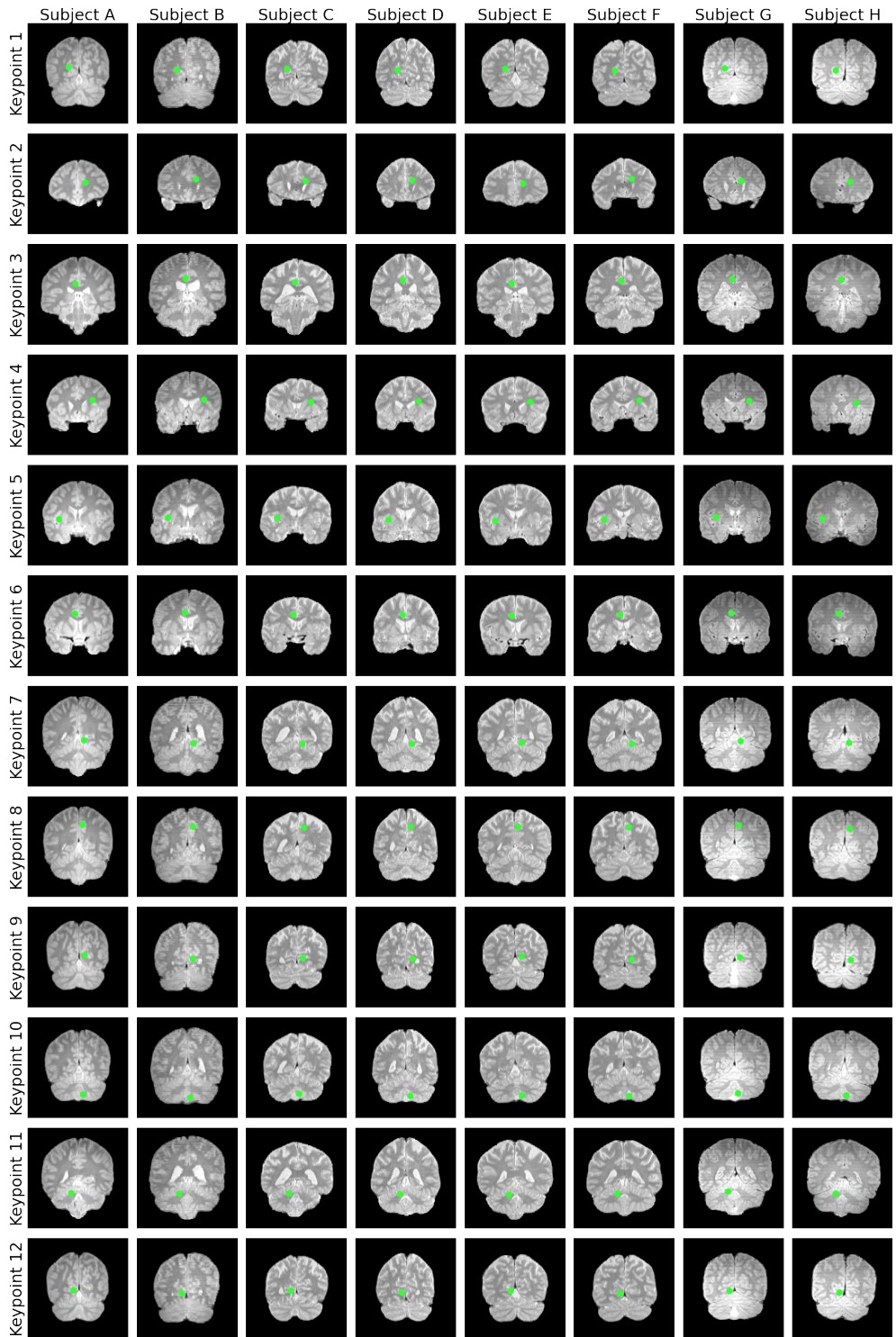

Figure A.7: Keypoints for different PD scans

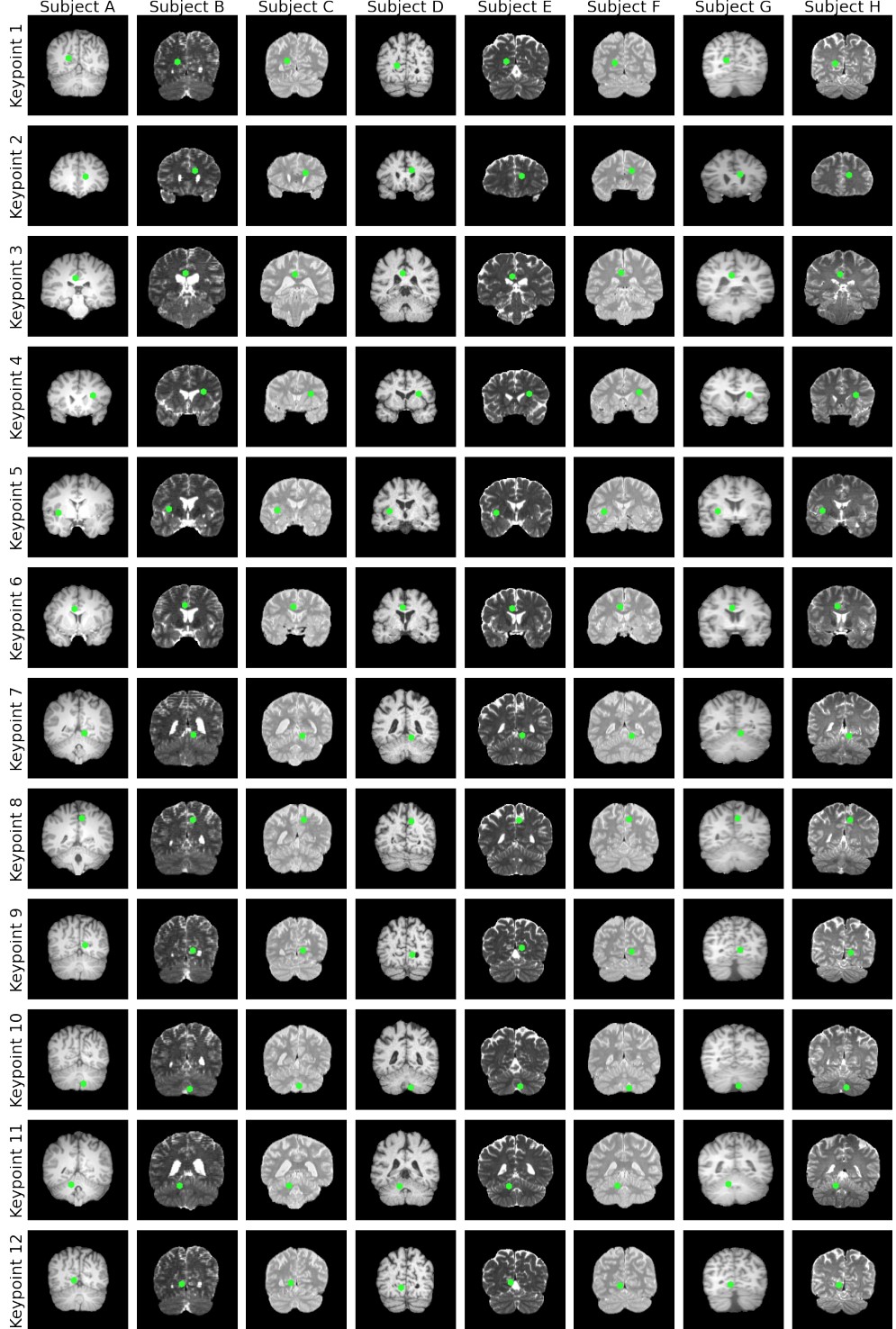

Figure A.8: Keypoints for different multimodal scans

