# OpenReview forum: "KeyMorph: Robust Multi-modal Affine Registration via Unsupervised Keypoint Detection"
_MIDL.io/2022/Conference — MIDL 2022_

### Official Review · Reviewer_Svyj · 2022-01-18

**Confidence:** 5
**Preliminary Rating:** 3
**Recommendation:** Poster

**Summary:**

The authors propose an unsupervised approach for the problem of image keypoint registration. They claim that the proposed approach is interpretable, robust to misalignment, and incorporates image symmetries.

The deep-learning part of the solution is supervised (to the best I could understand) and is in charge of finding the keypoints. This proposed approach is entirely depicted in Fig. 1 in the paper. To perform the experimental assessments, the authors used the IXI brain MRI pulic data set.  A comparison with two other methods is carried out.

**Strengths:**

- The paper is well written and mathematically soundy. I had no difficulties to follow the majority of the ideas of the work (although some ideas are passed sometimes in a confusing way);
- A very extensive survey of the literature, covering well recent works in the field of image registration;
- Appendices bring some interesting qualitative analysis.


**Weaknesses:**

Although the authors make some claims throughout the paper, it seems that some of them lack adequate motivations or rationales. This can be noted when, for example, the authors state that "the model is interpretable since the keypoints that drive the alignment can be visualized". Another example can be found at the end of page 3: "In contrast, our focus is on robust image registration via corresponding
keypoints; these prior works aim explicitly to obtain keypoints that are repeatable under different viewpoints and/or image acquisition conditions. Nevertheless, we build on previous ideas and introduce a framework that output matched keypoints in 3D space regardless of the initial position of the image." This is particularly confusing and questionable, and the authors miss to include a robust comparison with these other approaches or even motivate their statements.

I understood that the keypoints to be matched in the registration phase are found in a supervised way "Fixed and moving 3D images are passed through the same keypoint detection network composed of convolutional layers and a final center-of-mass (CoM) layer that predicts keypoints useful for registration", by using a CNN architecture. Only the registration stage is performed in an unsupervised fashion. Nevertheless, in Section 6, one can find something different: "a robust deep learning-based affine registration framework that employs unsupervised keypoint extraction". Did I miss something?

I think that the experimental analyses suffer from more comparative methods. The authors only used a traditional method and a Spatial Transformer Network. Considering Occam's razor, simpler methods could be considered for comparative analysis, such as the ones based on handcrafted features (like SIFT or SURF) and HOG descriptors, or even supervised CNN-based methods. Anyway, it seems to me that it is hard that the results could adequately support the initial claims in the present form.

It could be interesting to show some results of the matching part of the proposed method.





**Deanonymize Review:**

no

**Final Rating After The Rebuttal:**

3: Borderline

**Justification Of The Final Rating:**

I'm very happy to understand now the main ideas that were not clear in the first read of the paper. However, it seems to me that if the authors had included these explanations in the paper, it surely would be more self-contained. Especially the part of the supervision and the issue about scaling that was not clear in Fig. 2. I think this issue raised by one of the reviewers is important and should it be tackled, especially in Fig. 2.

As explanations were given but were not reflected in the text, I keep my rating in this paper.

**Paper Type:**

both

**Questions To Address In The Rebuttal:**

I would thank the authors to clear out the ideas about the training stage of the proposed approach. It was a little bit confusing to me.

Some qualitative analysis about keypoint registration also rested obscure, and it would be helpful to hear the authors about this.


**Special Issue:**

no

---

### Official Review · Reviewer_aM65 · 2022-01-23

**Confidence:** 4
**Preliminary Rating:** 4
**Recommendation:** Oral

**Summary:**

This paper introduces an approach to learn corresponding keypoints for use in multi-modal linear registration. Experiments were conducted on 3D brain MRI for T1, T2 and PD images, and the effectiveness of the approach in resolving synthetic transformations was demonstrated. The approach has clear benefits compared to optimisation based approaches and other learned approaches in terms of resolving large rotations and translations. Moreover, the extraction of keypoints provides an additional measure of interpretability. Given additional experimentation, this approach may indeed represent a useful step forward.

**Strengths:**

The use of learned keypoints for image registration has a clear benefit in terms of interpretability of the results, and explaining why a particular registration was chosen. Moreover, one could imagine a procedure where users select keypoints that they believe are meaningful to embed some amount of expert knowledge.

The other strength of this approach is that given keypoint locations, there are closed form approaches to derive globally optimal rigid, similarity of affine transformations.

The paper was clearly written, and figures 1 and 3 provide some nice illustrations of how the method works and inspecting the results.

**Weaknesses:**

One oddity was figure 2 does not consider a plot for changes in scale, only translation and rotation. One would anticipate that such an approach to learning keypoints may suffer more when considering changes in scale - so this is important to investigate. I could not see this discussed at any point in the paper.

As a colour blind reader, I found figure 2 very difficult to read without referring to the text/table in the appendix. I would suggest thicker lines in the legend and the use of different symbols for different methods.

It would also be useful to see how keypoint extraction (and associated alignment) is affected by noise/blurriness.

At testing time - are the different transformations (e.g. rotations/translations and scaling) applied separately or jointly? Given the description in appendix E, it looks like they are independently applied. It would be helpful to see quantitative results for images that have been translated, rotated and scaled.

**Deanonymize Review:**

no

**Detailed Comments:**

Given that you are only applying similarity transformations to the data - why are you not using the closed form estimation of a similarity transformation rather than an affine?

I think it would have been very helpful to show an illustration of what poor key points look like - to diagnose why they are unhelpful.

**Final Rating After The Rebuttal:**

5: Strong Accept

**Justification Of The Final Rating:**

the authors rebuttal answered my questions sufficiently, and some suggested revisions will be incorporated into the paper. The method and results are appealing enough to change my rating to strong acceptance.

**Paper Type:**

methodological development

**Questions To Address In The Rebuttal:**

The validation aspects described in the weaknesses should be addressed - particularly the changes of scale, as there is no justification given for its omission. It would also be helpful to understand the robustness to noise and realistic artefacts to fully demonstrate the benefits of this approach.

Figure 2 should also be modified to improve readability.

**Special Issue:**

no

---

### Official Review · Reviewer_s5Kr · 2022-01-25

**Confidence:** 4
**Preliminary Rating:** 5
**Recommendation:** Oral

**Summary:**

In the manuscript, the authors propose KeypointMorph, an unsupervised end-to-end learning-based image registration framework that relies on automatically detecting corresponding keypoints. Their core insight is that matching keypoints between images can be used to obtain the optimal transformation via a differentiable closed-form expression. They use this observation to drive the unsupervised learning of anatomically-consistent keypoints from images. They apply their method to the estimation of affine registration on multi-modal brain MRI scans. The obtained registration is robust to large misalignments and yields better interpretability since the keypoints reveal which parts of the image are driving the final alignment. They demonstrate registration accuracy that surpasses current state-of-the-art methods.

**Strengths:**


- Unsupervised end-to-end learning-based framework.
- Smart and novel approach.
- Robust to large misalignments.
- Interpretable.
- Good registration accuracy.
- The method can be designed to be equivariant under image translations and/or symmetric with respect to the input image ordering.
- The strategy leads to consistent keypoints, even across modalities.
- They have plans to make their code available.

**Weaknesses:**

- Table A.1. include also the time for training.
- Evaluation is somehow limited. It relies on the Dice coefficient.
- It is not clear what is the application for large misalignments.
- It is not evident how to extend the method to compute non-linear deformations.


**Deanonymize Review:**

no

**Final Rating After The Rebuttal:**

5: Strong Accept

**Justification Of The Final Rating:**

Nice proof of concept with great potential when nonrigid registrations would be included. Interesting for applications. The authors have addressed all my concerns. It would make a nice contribution to the conference.

**Paper Type:**

methodological development

**Questions To Address In The Rebuttal:**

- Include the time for training in Table A.1.
- Extend the quantitative evaluation to other metrics apart from the Dice coefficient. For example, manually placed landmarks.
- Clarify what the applications for large misalignments will be. When will be the method applicable in a clinical setting?
- Could you discuss how to extend the method to compute non-linear deformations?

**Special Issue:**

yes

---

### Meta-Review · Area_Chair_4RYv · 2022-02-18

**Recommendation:** Accept (Poster)
**Confidence:** 5

**Metareview:**

This paper presents a well-received novel idea: unsupervised keypoint detection for affine registration, including inter-modality image registration cases. The proposed method is practical to time-consuming and often less reliable cases with large misalignments. Reviewers’ critiques and suggestions were mainly about more clarifications of the evaluation part. They were well answered in the rebuttal, and authors are expected to include those modifications in the final manuscript as mentioned.

---

### Decision · Program_Chairs · 2022-02-28

Accept